# Antimicrobial and antibiofilm activity of human recombinant H1 histones against bacterial infections

Betsy Verónica Arévalo-Jaimes,[1,2] Mónica Salinas-Pena,[3] Inmaculada Ponte,[4] Albert Jordan,[3] Alicia Roque,[4] Eduard Torrents[1,2]

**ABSTRACT** Histones possess significant antimicrobial potential, yet their activity against biofilms remains underexplored. Moreover, concerns regarding adverse effects limit their clinical implementation. We investigated the antibacterial efficacy of human recombinant histone H1 subtypes against *Pseudomonas aeruginosa* PAO1, both planktonic and in biofilms. After the *in vitro* tests, toxicity and efficacy were assessed in a *P. aeruginosa* PAO1 infection model using *Galleria mellonella* larvae. Histones were also evaluated in combination with ciprofloxacin (Cpx) and gentamicin (Gm). Our results demonstrate antimicrobial activity of all three histones against *P. aeruginosa* PAO1, with H1.0 and H1.4 showing efficacy at lower concentrations. The bactericidal effect was associated with a mechanism of membrane disruption. *In vitro* studies using static and dynamic models showed that H1.4 had antibiofilm potential by reducing cell biomass. Neither H1.0 nor H1.4 showed toxicity in *G. mellonella* larvae, and both increased larvae survival when infected with *P. aeruginosa* PAO1. Although *in vitro* synergism was observed between ciprofloxacin and H1.0, no improvement over the antibiotic alone was noted *in vivo*. Differences in antibacterial and antibiofilm activity were attributed to sequence and structural variations among histone subtypes. Moreover, the efficacy of H1.0 and H1.4 was influenced by the presence and strength of the extracellular matrix. These findings suggest histones hold promise for combating acute and chronic infections caused by pathogens such as *P. aeruginosa*.

**IMPORTANCE** The constant increase of multidrug-resistant bacteria is a critical global concern. The inefficacy of current therapies to treat bacterial infections is attributed to multiple mechanisms of resistance, including the capacity to form biofilms. Therefore, the identification of novel and safe therapeutic strategies is imperative. This study confirms the antimicrobial potential of three histone H1 subtypes against both Gram-negative and Gram-positive bacteria. Furthermore, histones H1.0 and H1.4 demonstrated *in vivo* efficacy without associated toxicity in an acute infection model of *Pseudomonas aeruginosa* PAO1 in *Galleria mellonella* larvae. The bactericidal effect of these proteins also resulted in biomass reduction of *P. aeruginosa* PAO1 biofilms. Given the clinical significance of this opportunistic pathogen, our research provides a comprehensive initial evaluation of the efficacy, toxicity, and mechanism of action of a potential new therapeutic approach against acute and chronic bacterial infections.

**KEYWORDS** biofilm, *Galleria mellonella*, treatment, toxicity, proteins, antimicrobial peptides

**Peer Reviewer** Mujeeb Ur Rehman, Sichuan Agricultural University, Chengdu, Sichuan, China

Address correspondence to Eduard Torrents, etorrents@ibecbarcelona.eu, eduard.torrents@ub.edu.

The authors declare no conflict of interest.

See the funding table on p. 16.

Bacterial infections caused by multidrug-resistant bacteria are increasingly burdening the global healthcare system (1, 2). Bacteria have evolved various mechanisms to counteract the harmful effects of antibiotic molecules, including restriction drug uptake, upregulating efflux pumps, altering drug targets, and inactivating drugs (2). Additionally, bacteria have developed defense strategies at the multicellular level. Bacterial biofilms

are communities of bacteria growing together and working collectively to withstand threats that would kill planktonic cells (3). Cells in biofilms produce an extracellular matrix (ECM) that surrounds and protects bacteria against antimicrobials, the immune system, phages, and other attacks (3). The biofilm structure and ECM create diffusion gradients of nutrients and antimicrobial molecules, promoting metabolic adaptations that result in population heterogeneity, adding more complexity to biofilm eradication (3, 4). Consequently, biofilm-associated infections are clinically challenging and often become persistent and difficult to treat (5).

*Pseudomonas aeruginosa* is one of the six highly drug-resistant bacteria worldwide (2). Its metabolic adaptability, coupled with an arsenal of virulence factors, explains its prevalence in life-threatening acute and chronic infections, particularly among immunocompromised patients (6). Its ability to form biofilms with increased tolerance to antibiotics leads to the development of chronic wounds and pulmonary failure in patients with cystic fibrosis and other respiratory diseases (6, 7).

The urgent need to develop new antibiofilm strategies that can either replace or complement current alternatives has led to the identification of effective natural compounds, including phytochemicals, biosurfactants, and antimicrobial peptides (AMPs) (5). AMPs are an important part of the innate immunity of multiple species, and acquiring resistance to them is more complicated than to antibiotics (4, 8).

Histones are crucial proteins in eukaryotic DNA organization and regulation (9). However, they can be secreted passively (necrosis) or actively (apoptosis and neutrophil NETosis) into the extracellular fluid (9, 10). Their cationic characteristics lead to cell membrane damage upon interacting with phospholipids (9). Like AMPs, the antimicrobial potential of histones has been demonstrated both *in vitro* and *in vivo* by effectively killing bacteria, fungi, parasites, and viruses (10, 11).

Histone H1 is a basic protein that regulates the higher-order structure of chromatin. It has three structural domains: the N-terminal domain (NTD), the globular domain (GD), and the C-terminal domain (CTD). The CTD represents approximately half of the protein and has the highest content of basic residues (12). Human somatic cells can express up to seven H1 subtypes: H1.0-H1.5 and H1X (13, 14). Histone H1 was proposed as a component of the antimicrobial defense in the human gastrointestinal tract in the 90s (15, 16). Since then, its antimicrobial activity has been reported against various bacteria (10, 11, 17); however, its potential as antibiofilm therapy and *in vivo* efficacy remains largely unexplored.

This study aims to evaluate the activity of three human recombinant histone H1 subtypes against *P. aeruginosa* PAO1 planktonic and biofilm-growing cells. We selected two of the more abundant subtypes in mammalian cells, H1.2 and H1.4, and the subtype that increases in differentiated cells, H1.0 (full length and CTD). Furthermore, we evaluated their potential as combined treatment with antibiotics (ciprofloxacin [Cpx] and gentamicin [Gm]) and assessed their toxicity and efficacy *in vivo* using the invertebrate animal model *Galleria mellonella*.

## RESULTS

### Antimicrobial efficacy and synergy assessment

The antimicrobial efficacy of histones H1.0, H1.0 C-terminal domain (CTD), H1.2, and H1.4 was determined by measuring their minimal inhibitory concentration 50% ($MIC_{50}$) (18) in the bacterial growth of three different important pathogens (Table 1). Overall, higher antimicrobial activity was observed against both Gram-negative bacteria, *P. aeruginosa* PAO1 and *Escherichia coli* CFT073, with the first being more susceptible. Conversely, a minor effect was observed against the Gram-positive bacteria *Staphylococcus aureus*.

The biomass reduction of *P. aeruginosa* PAO1 planktonic cultures after 16 h of histone treatment at $MIC_{50}$ (Table 1) was visualized and measured by live and dead staining (Fig. 1). A statistically significant decrease in bacterial cell number (58%–70%) was shown in all samples treated with histone H1 subtypes compared to the control (Fig. 1A and B). Furthermore, some bacteria displayed red staining, indicating cell death (compromised

**TABLE 1** Antibacterial activity of the different histones

| Histone | $MIC_{50}{}^{a}$ | | |
|---|---|---|---|
| | *P. aeruginosa* PAO1 | *E. coli* CFT073 | *S. aureus* |
| H1.0 | 55 µg/mL (2.5 µM) | 154 µg/mL (7 µM) | >220 µg/mL[b] (>10 µM) |
| H1.0 C-ter | 58 µg/mL (5 µM) | >116 µg/mL[b] (>10 µM) | >116 µg/mL[b] (>10 µM) |
| H1.2 | 111 µg/mL (5 µM) | 222 µg/mL (10 µM) | >222 µg/mL[b] (>10 µM) |
| H1.4 | 46 µg/mL (2 µM) | 98 µg/mL (4 µM) | >229 µg/mL[b] (>10 µM) |

[a]The minimal inhibitory concentration 50% ($MIC_{50}$) values measured after 10 h of histones treatment are shown. Data are representative of three independent experiments.
[b]No bacterial growth inhibition was observed below the indicated concentration.

membranes). Thus, we calculated the percentage of viable cells (green stained) present in each group (Fig. 1C) and found a statistically significant reduction in samples treated with H1.4 (31%), H1.0 (27%), and H1.2 (18%) compared to the control.

Consequently, we explored the potential use of histones H1.0 and H1.4 combined with antibiotics. We evaluated the synergy with Cpx and Gm to inhibit the planktonic growth of PAO1. The results showed a synergistic effect only when histone H1.0 was combined with Cpx (Table 2), whereas the other combinations showed an additive effect.

## Histone and bacterial cells interaction

Aiming to elucidate the mechanism of action behind the antimicrobial activity of the H1 histones, we evaluated their direct interaction with *P. aeruginosa* PAO1 cells. Bacterial cultures were treated with histones for 30 min followed by a centrifugation step. Subsequently, we evaluated if histones were present in the supernatant or pellet fraction (Fig. 2). A precipitation control of histones without bacteria and a negative control of bacteria without histones were included. In Fig. 2A, it is observed that irrespective of the histone, if bacteria were absent, Western blot bands are observed in both pellet and supernatant. However, when mixed with bacteria, histones completely co-precipitate with them, suggesting a direct interaction of H1 histones with bacterial cells. Western blot bands were confirmed by immunoblot and post-transfer Coomassie staining (Fig. 2B), except in the case of H1.0 CTD due to unavailability of specific antibodies. Notably, upon incubation with *P. aeruginosa* PAO1, H1 histones appeared to undergo partial cleavage, producing smaller H1 forms, as indicated by the presence of multiple bands detected in the pellet by specific antibodies in the immunoblot (Fig. 2B). This suggests that H1 cleavage occurs due to interaction with *P. aeruginosa* PAO, likely mediated by secreted bacterial proteases. The cleaved form represents approximately 10%–40% of the total H1, depending on the H1 variant.

Furthermore, to assess if histone interactions caused any alteration to the bacterial cell membrane, cultures were treated with histones H1 at their *P. aeruginosa* PAO1 $MIC_{50}$ concentration (Table 1) and were stained with the specific dyes FM-464 and 4′,6-diami-dino-2-phenylindole (DAPI) (Fig. 3). Some cells with membrane damage are observed in the control samples after the centrifugation step (Fig. 3, control, zoomed images). However, very evident gaps in cell membranes were observed in all samples treated with histones (Fig. 3, histones, zoomed images) in a proportion of 5%–8% (cells with gaps in membrane/total cells). Additionally, histone H1.0 (full length and CTD) and H1.4 caused non-homogenous staining among cells, with most showing a fainter membrane compared to the control sample. This effect is less remarkable in samples treated with H1.2.

The differences observed in the alteration of bacterial membrane among histone subtypes could be explained by sequence and structural differences. Therefore, we analyzed several sequence-related properties of histone H1 subtypes, finding distinct features in the two subtypes with higher antimicrobial activity. Histone H1.0 has the highest positive charge density and the lowest hydropathicity, expressed as the GRAVY index, whereas H1.4 has the highest number of basic residues per protein molecule and the highest net charge (Table 3).

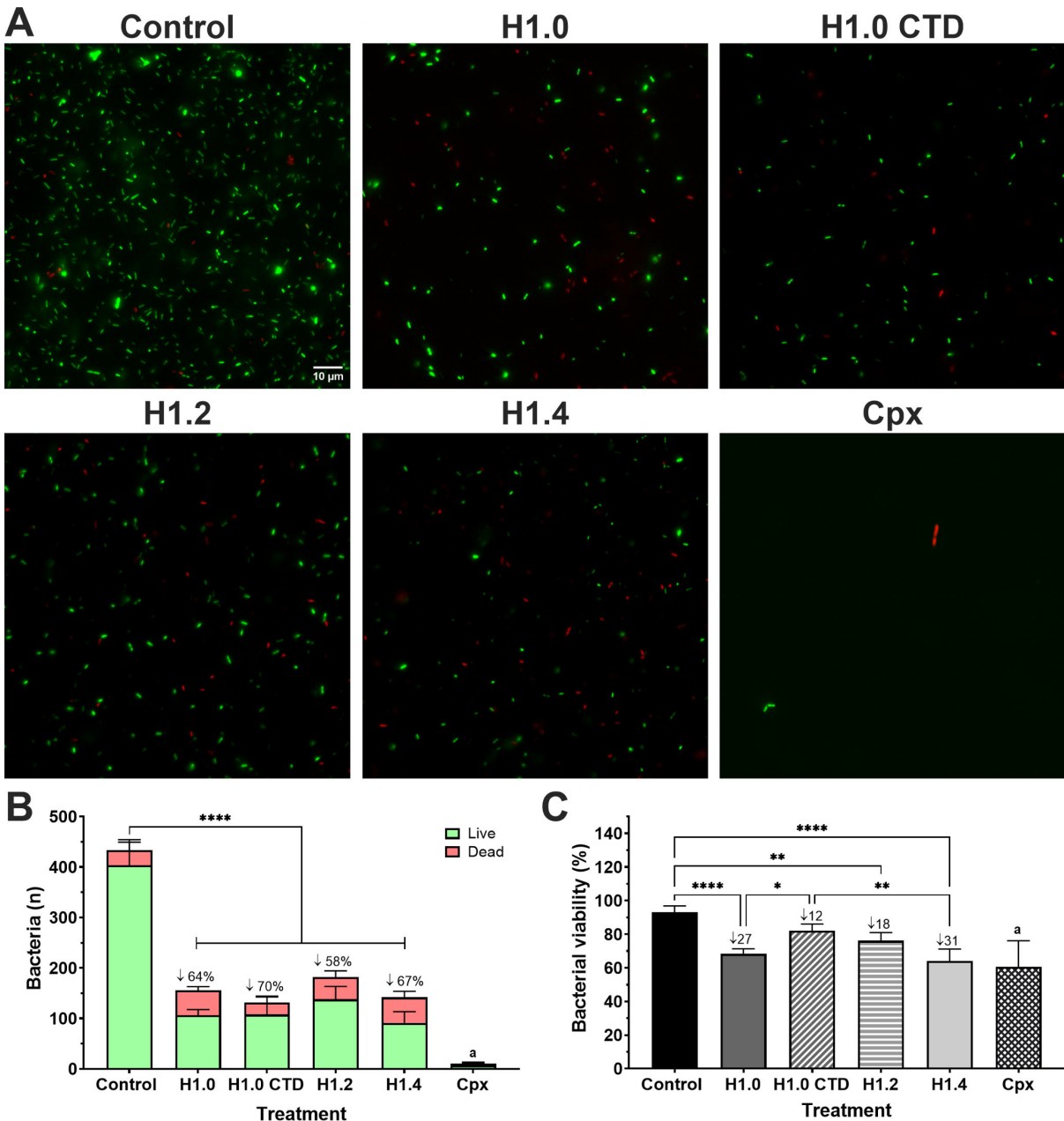

**FIG 1** Bacterial viability assessment of *P. aeruginosa* PAO1 planktonic cultures after H1 histone treatment. (A) Live and dead staining using Syto 9 (green live cells) and propidium iodide (red dead cells) dyes after 16 h of treatment. Histones were used at the $MIC_{50}$ concentration of PAO1: H1.0 at 55 µg/mL, H1.0 CTD 58 µg/mL, H1.2 at 111 µg/mL, and H1.4 at 46 µg/mL. Cpx was used at 2 µg/mL. Fluorescence images were processed with Image J. The scale bar of 10 µm is consistent for all cases. (B) Quantification of live and dead bacteria by particle counting with Image J. Numbers after ↓ symbol indicate the percentage of reduction in bacterial total cell number compared to the untreated control. Error bars display mean and standard deviation from at least four replicates. Differences in the number of total cells among groups were analyzed by a one-way ANOVA analysis with Tukey's multiple comparison test (****, *P* value < 0.0001). Letter a represents a statistical difference of total cell number in the Cpx group compared to the control, H1.0 and H1.2 with a *P* value < 0.0001, and H1.0 CTD and H1.4 with a *P* value < 0.001. (C) Percentage of viability of bacterial cells {[live bacteria/(live bacteria + dead bacteria)] × 100}, where live bacteria is the number of particles counted in the green channel (Syto 9 stained) and dead bacteria is the number of particles counted in the red channel (propidium iodide stained). Numbers after ↓ symbol indicate the percentage of reduction compared to the control. Error bars display mean and standard deviation from at least four replicates. Differences among groups were analyzed by a one-way ANOVA analysis with Tukey's multiple comparison test (*, *P* value < 0.05; **, *P* value < 0.01; ****, *P* value < 0.0001). Letter a represents a statistical difference in the viability percentage of the Cpx group compared to the control with a *P* value < 0.0001, H1.0 CTD with a *P* value < 0.001, and H.12 with a *P* value < 0.05.

**TABLE 2** Evaluation of synergy between histones H1.0 and H1.4 with ciprofloxacin and gentamicin on *P. aeruginosa* PAO1 growth[a]

| Combination | FIC$_{Histone}$ | FIC$_{Antibiotic}$ | ΣFIC | Interaction |
|---|---|---|---|---|
| H1.0 + Cpx | 0.25 | 0.25 | 0.5 | Synergy |
| H1.0 + Gm | 0.061 | 0.5 | 0.561 | Additivity |
| H1.4 + Cpx | 0.016 | 0.5 | 0.516 | Additivity |
| H1.4 + Gm | 0.012 | 0.5 | 0.512 | Additivity |

[a]FIC, fractional inhibitory concentration; Σ, summatory; Cpx, ciprofloxacin; Gm, gentamicin.

Moreover, we studied the secondary structure of the recombinant proteins by bioinformatic analysis and experimental methods. Sequence predictions showed the highest values of α-helix in H1.4 followed by H1.0 and H1.2 (Table 3). The helical content and induction were studied by circular dichroism (Fig. S1). We calculated the ratio between the molar ellipticity in the minimum of the α-helix canonical spectrum at 222 nm and the minimum of the entire spectrum in the random coil region as an index of the proportion of α-helix present in each sample. The proteins were analyzed in aqueous solution, where the spectrum is clearly dominated by the random coil and in the presence of trifluoroethanol (TFE), a known stabilizer of α-helix (19). We observed that, in contrast with the predictions, the molar ellipticity ratio was higher in H1.0 than in H1.4 and H1.2 in both conditions (Table 4).

To evaluate the existence of a correlation in the variation of the parameters of positive charge, hydropathicity, and α-helix content and the differences observed in the activity of histone H1 subtypes, we performed simple linear regressions. As a result, any correlation with the antimicrobial activity of histones was found (reduction of planktonic cells after treatment). However, there is a positive correlation between the proportion of α-helix content induced in TFE and the antibiofilm activity of the histone H1 subtype against static biofilms of PAO1 ($r^2 = 1$, $P$ value = 0.0043). Unfortunately, the low number of evaluated proteins made it impossible to perform a multiple linear regression that evaluates the effect of the parameters simultaneously.

## Histone H1 antibiofilm efficacy

Recognizing the significance of biofilm-related infections, we investigated the efficacy of histones against *P. aeruginosa* PAO1 biofilms. Initially, we assessed the reduction in biofilm biomass after treatment with H1 histones using a static biofilm model (96-well plate) and the crystal violet biomass staining. As shown in Figure 4A, all histone-treated samples exhibited some degree of biomass reduction, with more pronounced effects observed when using H1.0 (28%, adjusted $P$ value = 0.0273), followed by H1.4 (22%) and H1.2 (20%).

We selected H1.0 and H1.4 subtypes to test their antibiofilm *in vitro* efficacy in a continuous flow chamber biofilm assay, a model that better resembles *in vivo* biofilm infections. PAO1 biofilms treated with 100 µg/mL of each H1 subtype were dyed with the live and dead staining and imaged by confocal microscopy (Fig. 4B). As seen in Figure 4B, the orthogonal views show a significant decrease in bacterial biomass in the biofilm treated with histone H1.4 compared to the untreated control. Quantification confirmed a significant reduction in PAO1 biofilm biomass (29%) and average thickness (12%) after H1.4 treatment (Fig. 4C and D). Moreover, the dead biomass (red staining) significantly increased in the biofilm treated with H1.4 (Fig. 4E). No significant effect in the evaluated parameters was observed after H1.0 treatment. The findings obtained under this grown condition are particularly noteworthy because the resulting biofilms are highly resistant to antibiotics as seen with the Cpx treatment as positive control.

## *In vivo* H1.0 and H1.4 toxicity assessment and antimicrobial efficacy

Next, we used the *G. mellonella* larvae infection model to evaluate the antimicrobial potential of histone H1. No mortality or morbidity parameters (myelinization, activity reduction, and absence of cocoon) (20) associated with toxic effects were observed in *G.*

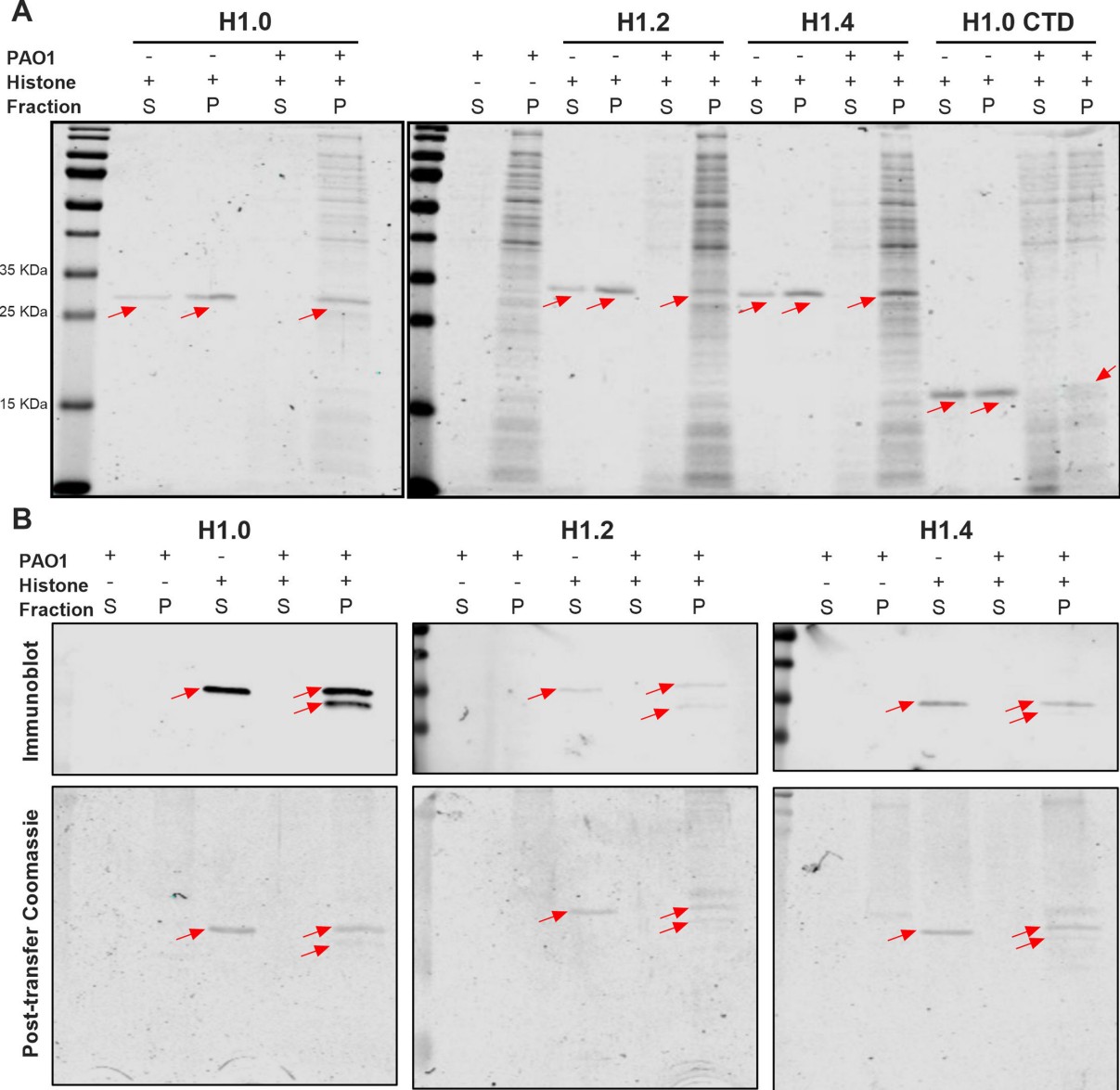

**FIG 2** H1 histones precipitation assay after incubation of PAO1 planktonic cells. (A) Coomassie-stained SDS-PAGE gel of supernatant and pellet fraction from bacterial cultures, histones, and bacterial cultures treated with histones after centrifugation. Red arrows indicate the band corresponding to the respective histone subtype. (B) Immunoblot and post-transfer Coomassie staining of bands found in the supernatant and pellet fraction from bacterial cultures, histones, and bacterial cultures treated with histones after centrifugation. +, presence; −, absence; S, supernatant; P, pellet.

*mellonella* larvae for up to 60 h post-injection with H1.0 and H1.4 histones in a range of 1.25–20 mg/kg (25–400 µg/mL), indicating low toxicity and a median lethal dose (LD$_{50}$) higher than 20 mg/kg. In addition, the lack of morbidity or mortality in the double-injected group (phosphate-buffered saline [PBS] followed by histones at 20 mg/kg) indicated no adverse effects when histones were administered during the treatment phase of a double injection procedure.

Therefore, we proceeded to evaluate the use of histones H1.0 and H1.4 as treatments for rescuing larvae infected with *P. aeruginosa* PAO1. At 1 h post-infection, larvae were treated with histones (100 µg/mL) alone and in combination with Cpx (Fig. 5). Larval survival was increased and extended when infected *G. mellonella* were treated with histones compared to those treated with PBS (negative control). Specifically, the median survival was significantly extended from 16 h in PBS-treated group to 18 and 20 h

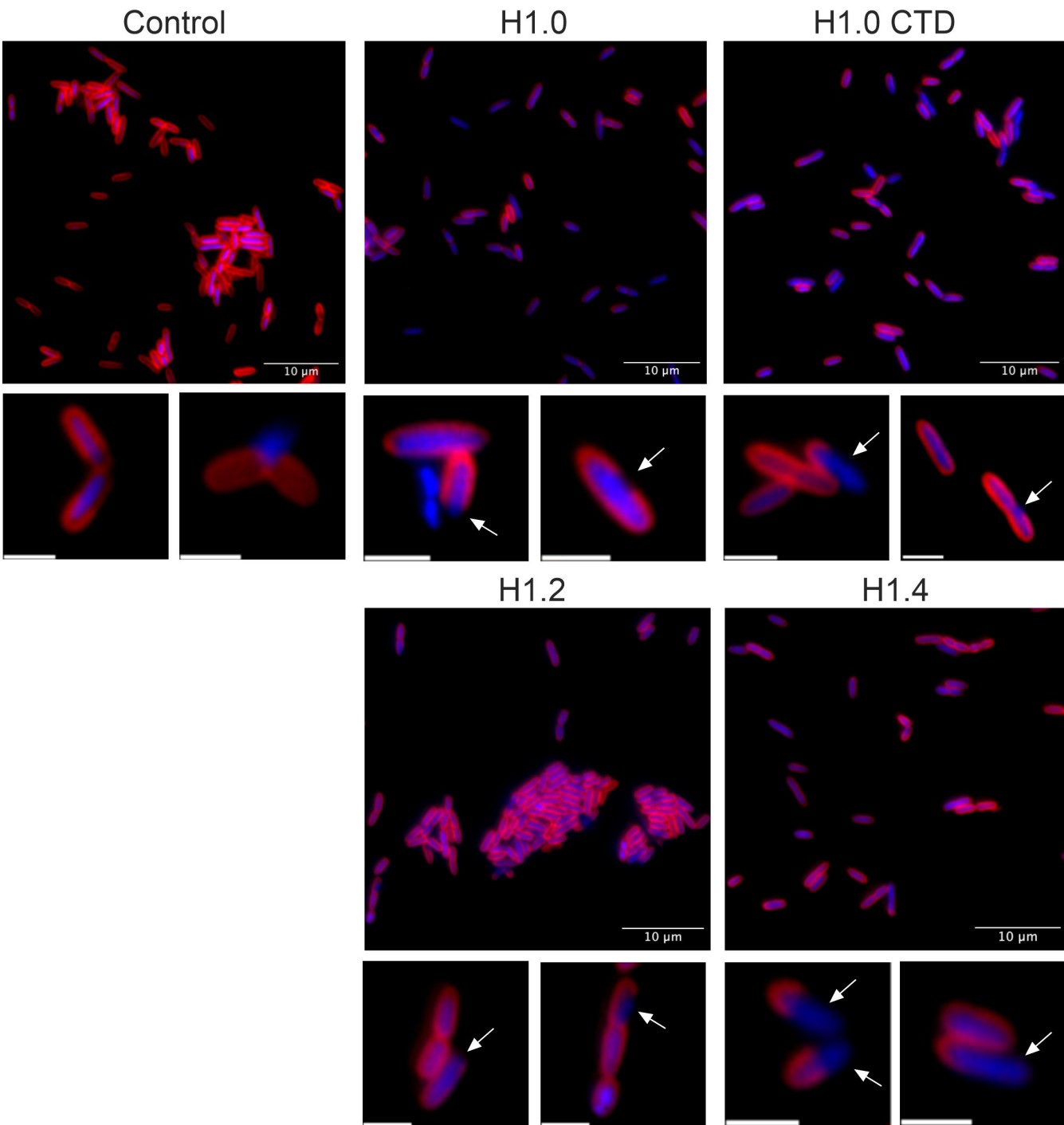

**FIG 3** Assessment of membrane alterations of PAO1 cells after histone treatment. Bacteria were stained with FM-464 (red) and DAPI (blue) dyes to visualize the membrane and genetic material, respectively. Histones were used at the $MIC_{50}$ of PAO1. Fluorescence images were processed with Image J software. Scale bars correspond to 10 µm in large images and 2 µm in zoomed images. White arrows point to gaps in cell membranes.

in the histone H1.0- and H1.4-treated groups, respectively. However, when comparing larvae treated with Cpx 1.5 mg/kg versus those treated with the combined therapy, no significant improvement was observed. Finally, the lack of mortality in the PBS–PBS group confirmed that the observed larval mortality in the other groups was related to bacterial infection rather than injuries associated with the double injection process.

**TABLE 3** Protein characteristics of H1 subtypes

| Subtype | Number of basic residues | Net charge | Positive charge density | GRAVY index | Secondary structure prediction (%) | | | |
|---------|--------------------------|------------|-------------------------|-------------|-----------|-----------------|-------------|-------------------|
| | | | | | α-Helix | Extended strand | Random coil | Ambiguous states |
| H1.0 | 62 | +53 | 0.32 | −1.07 | 35.57 | 4.12 | 50 | 10.31 |
| H1.2 | 62 | +55 | 0.29 | −0.68 | 32.39 | 2.35 | 55.87 | 9.39 |
| H1.4 | 66 | +59 | 0.30 | −0.78 | 38.81 | 2.28 | 46.12 | 12.79 |

## DISCUSSION

Histones are ubiquitous across different tissues, and their antimicrobial activity had been reported in multiple species (10). Moreover, some peptide fragments derived from histones cleavage are considered AMPs (21). For instance, H1 histones secreted by human colonic epithelial cells possess antimicrobial activity (15, 22), as well as the peptides derived from the NTD of H1 histone from Atlantic salmon and the CTD of H1 histone from rainbow trout (known as Oncorhyncin II) (23, 24).

In this work, we reported the ability of three human H1 subtypes (H1.0, H1.2, and H1.4) to inhibit bacterial proliferation and induce cell death in three well-known opportunistic pathogens: *P. aeruginosa*, *S. aureus*, and *E. coli* (Table 1; Fig. 1). Our findings support the study by Jacobsen et al. (25), where the efficacy of recombinant human histone H1.2 was demonstrated against bacteria found in human wound infections (25). Similarly, Jodoin and Hincke (17) reported that histone H1 from chicken (named H5 in this animal) has antimicrobial activity against Gram-negative and Gram-positive bacteria (17).

Our results (Table 1) confirm that histones had a broad-spectrum antimicrobial effect regardless of the subtype used. However, a lower concentration of the protein is required to inhibit Gram-negative bacteria (*P. aeruginosa* PAO1 and *E. coli* CFT073). This observation is consistent with the affinity of H1 for bacterial lipopolysaccharide (LPS), the major component of the outer membrane of Gram-negative bacteria (26). Our study confirms an interaction between H1 histones and *P. aeruginosa* PAO1 cells upon mixed incubation (Fig. 2). Moreover, the fragmentation of all three histone subtypes in the precipitate fraction, alongside bacteria, suggests that bacterial proteases may degrade the histones, and their antimicrobial properties could be attributed to AMP characteristics.

The mechanism underlying histone-mediated bacteria killing remains unclear, but two primary options have been proposed: membrane disruption or cytoplasmatic targeting. H1 from the liver of Atlantic salmon causes morphological alterations that directly damage the cell surface (27). Likewise, H1 from chickens caused pore formation and content liking in *P. aeruginosa* cells (17). Conversely, oncorhyncin II is unable to form stable channels in the membrane, suggesting an antimicrobial activity resultant from intracellular processes (24). In this study, bacterial cells treated with histone H1 subtypes displayed clear membrane gaps, indicating bacterial killing by alteration of membrane integrity (Fig. 3). Furthermore, we hypothesized that their presence in the bacterial membrane hindered FM-464 dye incorporation, resulting in less intense staining in histone-treated samples compared to the untreated control. This effect was more pronounced in the case of H1.0 and H1.4, suggesting a higher affinity of these histones with *P. aeruginosa* PAO1 membrane.

Several factors can influence how an AMP causes membrane alterations, including propensity for peptide self-assembly, net charge, and amphipathicity (21, 28). These characteristics can be found in peptides with an amphipathic α-helix structure (29, 30).

**TABLE 4** Circular dichroism of H1 subtypes

| Solvent | Molar ellipticity ratio[a] | | |
|---------|------|------|------|
| | H1.0 | H1.2 | H1.4 |
| PBS | 0.55 | 0.43 | 0.42 |
| TFE 20% | 0.63 | 0.56 | 0.57 |

[a]Molar ellipticity ratio was calculated as the value at 222 nm referred to the minimum of the spectrum.

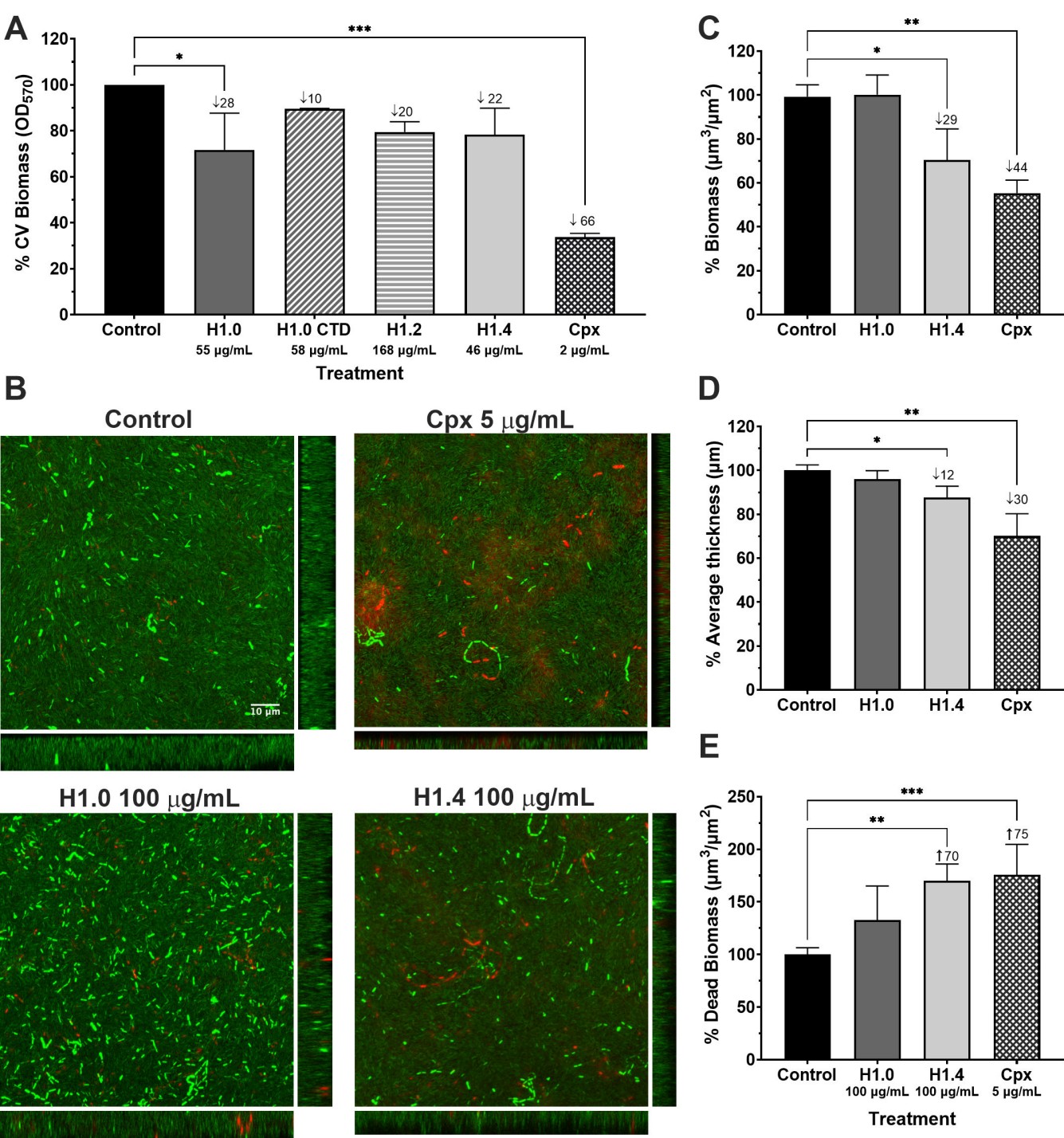

**FIG 4** Antibiofilm activity of histones against PAO1 biofilms. (A) Biomass reduction of static biofilms after histone treatment. Error bars display mean and standard deviation of at least three replicates. Data differences with respect to the control group were analyzed by a one-way ANOVA analysis with Dunnett's multiple comparison test (*, $P$ value < 0.05; ***, $P$ value < 0.001. (B) Confocal microscopy visualization of continuous flow biofilms after treatment with H1.0 and H1.4. Biofilms were stained with Syto 9 and propidium iodide to dye live cells in green and dead cells in red. Z-stacks from confocal images with their corresponding orthogonal views are displayed. The scale bar of 10 µm is consistent across all images. Confocal image analysis of the (C) percentage reduction in biomass, (D) percentage reduction in average thickness and (E) percentage increase in dead biomass after treatment of flow biofilms with H1.0 and H1.4. Numbers after the ↓ symbol indicate the percentage of reduction compared to the control, whereas numbers after the ↑ symbol indicate the percentage of increase compared to the control. Error bars display mean and standard deviation of at least three replicates. Data differences with respect to the control group were analyzed by a one-way ANOVA analysis with Šidák's multiple comparison test (*, $P$ value < 0.05; **, $P$ value < 0.01; ***, $P$ value < 0.001).

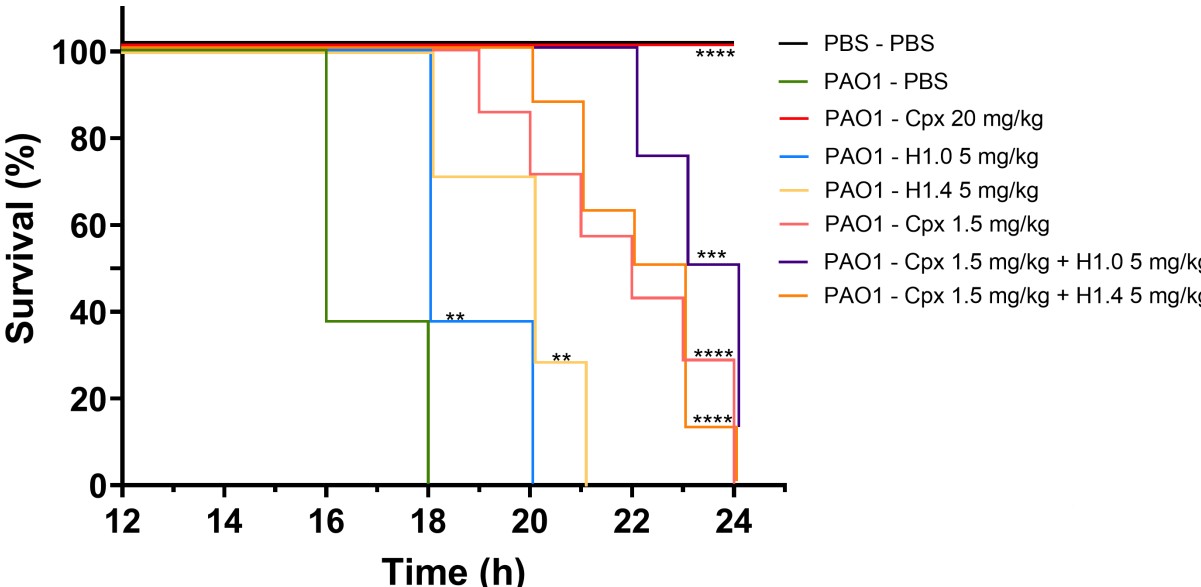

**FIG 5** *In vivo* evaluation of H1 histone activity against *G. mellonella* infected with PAO1. Kaplan-Meier survival curves of *G. mellonella* larvae infected with PAO1 and treated with histones alone or in combination with Cpx. Control larvae were injected with PBS. Larval mortality was monitored for 16–24 h post-injection with observations conducted hourly. Significant differences in treatment efficacy compared to the PAO1-PBS-treated group were assessed using a log-rank test (*, $P$ value < 0.05; **, $P$ value < 0.01; ***, $P$ value < 0.001; ****, $P$ value < 0.0001). The results depicted in this figure represent an experiment replicated three times with consistent outcomes, each condition involving eight larvae.

We compared the positive charges, hydropathicity, and α-helix content of H1.0, H1.2, and H1.4 proteins (Tables 3 and 4). Overall, H1.0 and H1.4, the subtypes with higher antimicrobial and antibiofilm activity, exhibited higher net charge, lower gravy index (more hydrophilic), and higher α-helix content. Simple linear regression models showed a correlation between α-helix content induced in TFE and the antibiofilm activity of H1 subtypes against *P. aeruginosa* PAO1 static biofilms. However, the structural properties of the three subtypes are very similar, and more robust statistical analysis need to be conducted.

As not all the antimicrobial molecules had antibiofilm activity, we decided to test H1 histones' efficacy against *P. aeruginosa* PAO1 biofilms (Fig. 4). The urgent need for alternative molecules that can eradicate biofilms, specially from multidrug-resistant bacteria, has led to the identification of several AMPs with antibiofilm properties (see the Biofilm-active AMPs database at http://www.baamps.it/) (31). However, as far as we are aware, the ability of histones for biofilm disruption has been understudied. Jacobsen et al. showed a reduction of ~60% of bacterial burden in an *in vivo* rat burn model infected with *P. aeruginosa* after treatment with H1.2 protein (25). Likewise, the work from Jodoin and Hincke (17) reported that the minimum biofilm eradication concentration (MBEC) value of H1 from chicken against *P. aeruginosa* biofilms formed under static conditions is >128 µg/mL (17). Although the methodology used in our study is different, our data pointed to a similar MBEC value.

We used two different *in vitro* models widely employed for biofilm studies. The microtiter plate, a static model used, is easy to use and has a high throughput; and the continuous flow chamber assay is a dynamic model in which the resulting biofilms are structurally similar to those present in natural infections (32). Flow conditions stimulate the production of robust ECM, which, in the case of *P. aeruginosa* PAO1 biofilms, contain alginate and extracellular DNA—two highly negatively charged molecules that can trap positively charged agents (33). Thus, we expected this scenario to be more challenging for histone treatment efficacy. We found higher antibiofilm activity in H1.0 in the static model compared to the other H1 subtypes (Fig. 4A). However, in the dynamic biofilm model, H1.4 has better results than H1.0 (Fig. 4B and D). H1.0 has a higher density of

positive charges, lower gravy index (more hydrophilic), and higher formation of α-helix in TFE compared to H1.4 (Table 3). Similarly, H1.0 CTD, the protein region with more positive charges and less hydrophobicity (12, 34), showed reduced activity against *P. aeruginosa* PAO1 biofilms (Fig. 4A) when compared to H1.0 (full length). This could indicate that these parameters are influencing the antibiofilm activity of histones, with their effect being more remarkable when biofilms are formed under dynamic conditions.

It is important to consider that the biofilm reduction assessed by the static crystal violet assay quantifies both cell and ECM biomass, whereas live and dead staining only assess differences in cell biomass. Nevertheless, we found the almost 30% reduction in cell biomass obtained after H1.4 treatment in the flow biofilm assay to be highly significant (Fig. 4B and C). Considering the reduction observed in the average biofilm thickness (Fig. 4D) and the increase in the biomass stained with propidium iodide (Fig. 4E), we proposed that the antibiofilm activity of the histone H1.4 is associated with its capacity to induce bacterial cell death.

Finally, the antimicrobial activity of H1 subtypes against *P. aeruginosa* PAO1 obtained in this study was validated in an *in vitro* animal model. It is known that the use of histones as therapeutic agents could lead to adverse effects. For instance, extracellular histones are damage-associated molecular patterns, able to induce chemokine release and cytotoxicity (10). They can induce thrombin generation and platelet aggregation, promoting thrombosis and thrombocytopenia (21). Moreover, histone interaction with cell membranes can activate adverse inflammatory responses that worsen several diseases (21).

Regarding histone H1, oncorhyncin II and H1 from chicken do not have significant hemolysis (17, 24). Similarly, human recombinant H1.2 showed very low hemolytic activity (2%) at 250 µg/mL, but cytotoxicity against human keratinocytes at low concentrations was observed (25). In this study, we did not observe any sign of toxicity in *G. mellonella* larvae injected with histone H1.0 and H1.4 up to 20 mg/kg (400 µg/mL). Furthermore, we found an increase of the median survival rate after treatment of *G. mellonella* infected with *P. aeruginosa* PAO1 (Fig. 5).

*G. mellonella* larvae have an innate immune response very similar to that found in mammals, making them widely used for initial testing of the efficacy and toxicity of new antimicrobial agents, thereby reducing the use of mammalian models (35). However, we are aware that further toxicity and efficacy studies involving human cells and mammals must be conducted, but the positive results obtained against this aggressive bacterium are promising. In addition, the discovery that polysialic acid can reduce the H1-mediated cytotoxicity in eukaryotic cells without altering its antimicrobial activity creates a new alternative in biomedical applications (9).

Beyond cytotoxicity, there are still significant challenges and limitations to overcome before histone-derived AMPs, and AMPs in general, can be used clinically (36, 37). These challenges include the high cost for large-scale manufacturing of natural peptides, susceptibility to proteolytic degradation, and low bioavailability (17, 36, 38). However, new strategies are continually being proposed (39, 40), including the use of shorter peptides derived from histone sequences, structural modifications (D-amino acids, cyclic peptides, acetylation, amidation, etc.), and delivery systems (liposomes, nanoparticles, micelles, hydrogels, etc.) to increase stability (17, 36, 38, 40). In terms of formulation, most AMPs designed for clinical use are limited to local administration or intravenous injection, although lipid-based nanoparticles have been proposed to overcome the challenges associated with oral administration (36).

Another strategy to enhance clinical potential involves the concomitant use of histones with AMPs that have different modes of action, which has been shown to be effective (41). Similarly, combining AMPs with antibiotics has been proven to enhance antibacterial activity (8). In this study, we tested the synergy between histones and two antibiotics with different mechanisms of action. First, we did not observe any improvement when combining histones with the aminoglycoside Gm. We believe Gm's

ability to interact with LPS (42) may create competition for histone–bacteria interactions, diminishing their antimicrobial potential.

Second, although we observed a synergy between Cpx and H1.0 *in vitro* (Table 2), no significant difference in the median survival rate of *G. mellonella* larvae was observed with the combined treatment compared to the antibiotic alone (Fig. 5). This discrepancy may be due to a masking of the histones' antibacterial effect by Cpx in the *in vivo* test. Further experiments should include a reduced Cpx concentration (<1.5 mg/kg) where all larvae die between 19 and 21 h, as observed after *P. aeruginosa* PAO1 infection and histone treatment (Fig. 5). This would allow us to determine whether there is an extension of larval survival with the combined strategy. We acknowledge this as a limitation of our study.

In conclusion, we have demonstrated that H1.0, H1.2, and H1.4 have antimicrobial activity against *P. aeruginosa* PAO1, likely attributable to membrane disruption. H1.0 and H1.4 were the subtypes that exhibit higher antimicrobial activity with lesser concentration required. H1.4 had higher antibiofilm potential and better survival outcomes *in vivo* in an acute infection model of *P. aeruginosa* PAO1. Although we did not validate a synergistic effect of H1.0 and H1.4 with the antibiotics ciprofloxacin and gentamicin, none of the histone subtypes caused toxicity to *G. mellonella* larvae.

## MATERIALS AND METHODS

### Histone expression and purification

H1.0 and its CTD were cloned into pQE60 (Qiagen). The H1.0 CTD was cloned directionally into the restriction sites Nco I and Bgl II of the plasmid vector. In the case of the full-length H1.0, the pQE-60 vector was digested with Nco I, then the DNA ends were blunted with the Klenow fragment of DNA polymerase I (Roche Diagnostics) and finally digested with Bgl II. The digested plasmid was ligated to the amplification product digested with Bgl II. Recombinant H1.0 and its CTD were expressed in *E. coli* M15 (Qiagen) (43). Histones H1.2 and H1.4 were cloned into pET21a using the same cloning strategy, inserting the coding sequence between Nde I and Xho I restriction sites. Expression of H1.2 and H1.4 was carried out in *E. coli* BL21 (DE3) (Merk Millipore).

All proteins contained a His-tag at the C-terminal protein end. Recombinant proteins were expressed and purified by metal-affinity chromatography as previously described (44). Cells were grown to an $OD_{600}$ of 0.8 and then induced with 1 mM isopropyl β-d-1-thiogalactopyranoside (IPTG), allowing expression to proceed for 4 h at 37°C. Cells were then harvested and stored at 80°C until use. Cells were lysed in lysis buffer (0.05 M $NaH_2PO_4$, 0.75 M NaCl, 0.02 M imidazole) plus 4 M guanidine hydrochloride, pH 8.0, for 15 min at room temperature. Guanidine hydrochloride was added to avoid degradation and aggregation of the expressed protein. The extract was centrifuged at 20,000 × *g* for 25 min. The supernatants were loaded on a HiTrap chelating HP column (Amersham Biosciences) equilibrated with lysis buffer. The column was washed in three steps with lysis buffer containing increasing amounts of imidazole: 40, 60, and 80 mM. Finally, the proteins were eluted with 250 mM imidazole in lysis buffer and desalted by gel filtration through Sephadex G-25 (Amersham Biosciences). Before their use, histones were thawed, vortexed, and sonicated (three cycles of 20 s) using an ultrasonic processor UP50H (Hielscher ultrasonics).

### Sequence analysis and predictions

The sequences of H1.0, H1.2, and H1.4 were used to compare the positive charge density and the predicted secondary structure of the analyzed subtypes. Positive charge density of H1.0 (P07305), H1.2 (P16403), and H1.4 (P10412) was calculated as the quotient between the number of basic residues and the protein length. The GRAVY index was calculated by the sum of hydropathy values of all amino acids divided by the protein

length. Consensus secondary structure predictions were obtained at the Network Protein Sequence @nalysis (NPS@) server (https://prabi.ibcp.fr/htm/site/web/app.php/home).

## Circular dichroism

Secondary structure of H1.0, H1.2, and H1.4 was analyzed in aqueous solution (PBS) and in the presence of 20% 2,2,2 trifluoroethanol. Proteins were resuspended in the desired solvent at a final concentration of 230 ng/µL. Measurements were recorded in a J-815 spectropolarimeter (JASCO) in a continuous scanning mode, in the 190–250 nm range at a scanning speed of 200 nm/min with a bandwidth of 1 nm. Five accumulations were taken of each spectrum. Molar ellipticity was obtained by normalizing circular dichroism (CD) units by the protein concentration and the number of residues.

## Bacterial strains and growing conditions

Three different bacterial strains were used in this study: *P. aeruginosa* PAO1 strain (CECT 4122, ATCC 15692), referred to as PAO1, *S. aureus* (CECT 86, ATCC 12600), and a uropathogenic *E. coli* CFT073 strain (ATCC 700928). Bacteria were retrieved from a −80°C stock and were cultured on Luria-Bertani media (Scharlau) for PAO1 and *E. coli* CFT073, and tryptic soy agar/broth (TSA/TSB) (Scharlau) for *S. aureus*. Overnight cultures were made by incubation at 37°C and 200 rpm. Unless specified, bacterial inoculum for each experiment was prepared by incubating 100 µL of overnight cultures in fresh TSB media until reaching the initial exponential log phase (OD$_{550}$ ≈ 0.2–0.3).

## Antibacterial activity against planktonic bacteria

One hundred microliters of inoculum from all three bacterial strains was plated in a 96-well microtiter plate (Corning) filled with histone protein concentrations ranging from 2 to 10 µM. TSB medium was used as a negative control. Following a previously described method (18), the microtiter plate was incubated at 37°C in a SPARK Multimode microplate reader (Tecan) with 150 rpm shaking. Bacterial growth was monitored for 10 h with readings taken every 15 min.

The MIC$_{50}$ was defined as the compound concentration that reduces bacterial growth, determined as the OD$_{550}$, by 50%.

## Fluorescent microscopy viability test analysis

Aliquots of 500 µL from a *P. aeruginosa* PAO1 inoculum were treated with histones at a concentration equal to their MIC$_{50}$ value of each subtype (see Table 1). Samples were treated with Cpx at 2 µg/mL, and media were included as controls. After 16 h of incubation at 37°C, 100 µL of the samples was centrifuged at 6,000 rpm for 5 min. Pellets were resuspended in 25 µL of PBS (Fisher Scientific S.L.) and stained with Syto 9 and propidium iodide (Live/Dead BacLight Bacterial Viability Kit, Thermo Fisher Scientific) following the manufacturer's instructions. Samples were visualized using a Nikon inverted fluorescent microscope ECLIPSE Ti–S/L100 (Nikon, Japan) coupled with a DS-Qi2 Nikon camera (Nikon, Japan). Images were processed and quantified by analysis of particles with ImageJ software. The percentage of viability was calculated according to the following formula:

$$\% \text{ viability} = \frac{\text{live bacteria}}{\text{live bacteria} + \text{dead bacteria}} \times 100$$

where live bacteria is the number of particles counted in the green channel (Syto 9 stained), and dead bacteria is the number of particles counted in the red channel (propidium iodide stained).

## Antibiotic-histone synergy testing by checkerboard assay

The synergy of histones H1.0 and H1.4 in combination with Cpx and Gm was determined by the standard broth microdilution assay as described previously (43). Briefly, *P. aeruginosa* PAO1 inoculum was added in a 96-well microtiter plate with increasing concentrations of histone on one axis (0.15–20 µM) and increasing concentrations of antibiotic on the other (0.01–2 µg/mL of Cpx or 0.125–2 µg/mL of Gm). Plates were incubated at 37°C in a SPARK Multimode microplate reader with 150 rpm shaking for 16 h. The effect of the antimicrobial combination was defined using the lowest fractional inhibitory concentration (FIC) index (45). The formula employed for FIC calculation (43) and its interpretation (46) were as previously described:

$$\Sigma\text{FIC} = \frac{\text{MIC}_{100}\ \text{combination}}{\text{MIC}_{100}\ \text{histone}} + \frac{\text{MIC}_{100}\ \text{combination}}{\text{MIC}_{100}\ \text{antibiotic}},$$

where $\Sigma\text{FIC} \leq 0.5$ is defined as synergy, $\Sigma\text{FIC} > 0.5$ and $< 4$ is additivity/indifference, and $\Sigma\text{FIC} > 4$ is antagonism.

## Histone and bacterial cell interaction by precipitation experiments

Aliquots of 200 µL of a *P. aeruginosa* PAO1 inoculum were treated with a concentration of 10 µg/mL of each histone subtype. After 30 min of incubation at 37°C and 300 rpm in a thermomixer (Eppendorf), samples were centrifuged at 6,000 rpm for 5 min, and supernatant and pellet were collected. Histones added to TSB were used as precipitation control.

Subsequently, supernatant and pellet samples were boiled (5 min at 95°C) and exposed to SDS-PAGE (14%), followed by Coomassie staining or immunoblotting with H1 variant-specific antibodies. For immunoblotting, protein extracts were transferred to polyvinylidene fluoride (PVDF) membrane, blocked with 5% non-fat milk for 1 h, and incubated overnight at 4°C with the primary antibodies anti-H1.0 (Millipore 05-629l), anti-H1.2 (Abcam ab4086), and anti-H1.4 (Invitrogen 702876). No specific antibodies against the C-terminal region of H1.0 were available. Next, samples were incubated with secondary antibodies conjugated to fluorescence (IRDye 680 goat anti-rabbit IgG or IRDye 800 goat anti-mouse IgG, Li-Cor) for 1 h at room temperature. Bands were visualized using an Odyssey Infrared Imaging System (Li-Cor). Coomassie staining was used as a loading control.

## Bacterial membrane integrity after H1 incubation

Aliquots of 200 µL of a *P. aeruginosa* PAO1 inoculum were treated with histones H1.0, H1.0 CTD, H1.2, and H1.4 at a concentration of 55, 58, 111, and 46 µg/mL, respectively, which correspond to their MIC$_{50}$ value (Table 1). After 30 min of incubation at 37°C and 300 rpm in a thermomixer (Eppendorf), samples were centrifuged at 6,000 rpm for 5 min. Pellets were resuspended in 50 µL of PBS with DAPI (Invitrogen) at 100 µg/mL and FM 4-64 (Invitrogen) at 40 µg/mL for 15 min. Samples were dropped on a slide pre-covered with agarose 1% (wt/vol) as previously described (47) and visualized with a Nikon inverted fluorescent microscope ECLIPSE Ti-S/L100 (Nikon, Japan) coupled with a DS-Qi2 Nikon camera (Nikon, Japan). Images were processed with Image J software. The percentage of cells with membrane gaps was calculated by manually counting damaged cells and dividing by the total number of cells (automated particle counting) from three different replicates, using images with a 63× magnification and ImageJ software.

## Antibacterial efficacy against biofilms

For static biofilm analysis, an overnight culture from *P. aeruginosa* PAO1 was diluted to an OD$_{550}$ = 0.1 in TSB supplemented with 0.2% glucose (Fisher Scientific S.L.), added in a 96-well microtiter plate, and incubated at 37°C. After 72 h, wells were washed three

times with PBS and treated with the different H1 histones at PAO1 $MIC_{50}$ (Table 1). TBS and Cpx at 2 µg/mL-treated wells were used as controls. Six hours after treatment, three PBS washes were carried out, and wells were fixed with methanol (Fisher Scientific) for 15 min. Then, 1% (wt/vol) Crystal violet (Merck Life Science) was added for 5 min, followed by a distaining step with 30% acetic acid (Scharlau). Biofilm biomass was determined by measuring the absorbance ($OD_{570}$) using a Microplate spectrophotometer Benchmark Plus (Bio-Rad, USA).

The antibiofilm efficacy of histones H1.0 and H1.4 was also tested in a continuous PAO1 flow biofilm assay as previously described (18). Briefly, an overnight culture diluted at $OD_{550} = 1$ was inoculated into a flow chamber and was allowed to attach for 2 h. Then, fresh TSB + 0.2% glucose was continuously pumped (42 µL/min) through the flow chamber for 72 h until a mature biofilm was obtained. The mature biofilms were treated with histones at 100 µg/mL and left to act for 6 h. Treatments with TBS and Cpx at 5 µg/mL were used as controls. Finally, biofilms were stained with Syto 9 and propidium iodide according to the manufacturer's instructions and were visualized with a Zeiss LSM 800 confocal laser scanning microscope (CLSM). Image processing was performed with Image J software, and measurements of biofilm biomass (green channel), average thickness (green channel), and dead biomass (red channel) were obtained using FIJI and COMSTAT2 plugins (48).

## Antibacterial efficacy and toxicology in a *G. mellonella* infection model

*G. mellonella* larvae were fed with an artificial diet (15% corn flour, 15% wheat flour, 15% infant cereal, 11% powdered milk, 6% brewer's yeast, 25% honey, and 13% glycerol) and maintained at 34°C in the dark (49). Larvae selected for all experiments had a weight range of 175–250 mg.

First, the toxicity of H1.0 and H1.4 histones was evaluated by injecting 10 µL of five different concentrations into the top right proleg of the larvae using a microsyringe (Hamilton): 1.25, 2.5, 5, 10, and 20 mg/kg. Considering that 2.5 mg/kg is equivalent to 50 µg/mL, the evaluated concentration range comprises ~0.5× $MIC_{50}$ to ~8× $MIC_{50}$ of *P. aeruginosa* PAO1 to H1 histone subtype (Table 1). PBS-treated larvae were included as negative control. Because testing histone activity against infected larvae involved a double injection protocol (infection and treatment), histone toxicity was also tested in previously injected larvae. To this end, larvae injected with PBS were treated 1 h later with 10 µL of histones at 20 mg/kg though the top left proleg. The experiment was conducted twice and included a total of five larvae per group each time. Larvae were monitored for 60 h.

Then, the antimicrobial efficacy of histones against *P. aeruginosa* PAO1 infection in *G. mellonella* was tested. Larvae were infected with *P. aeruginosa* PAO1 (5–50 CFUs/larva) and 1 h after, 5 mg/kg of histone H1.0 and H1.4 was injected, alone and in combination with 1.5 mg/kg of Cpx. Infected larvae were also treated with PBS as a negative treatment control, Cpx 20 mg/kg as a positive control of treatment, and Cpx 1.5 mg/kg as a control of synergy effect between antibiotic and histones. Additionally, a group of larvae injected with PBS at both the infection and treatment steps was included as a control for the double injection, to rule out mortality due to consecutive injuries from the injection process. The experiment was conducted three times with eight larvae per group. Larvae were monitored from 16 to 24 h post-infection, with observations every hour.

## Statistical analysis

All data were statistically analyzed using GraphPad Prism version 10.00 (GraphPad Software, USA). Normality was assessed using the Shapiro-Wilk test. Comparison of means among groups was performed using one-way analysis of variance (ANOVA) with corrections for multiple comparisons (Tukey, Dunnett, or Šidák), as specified in each figure legend. Correlations among structural parameters and the antimicrobial and antibiofilm effect of histone H1 subtypes were evaluated using simple linear regressions.

Comparison of Kaplan-Meier survival curves was made by log-rank tests. A $P$ value of <0.05 was considered statistically significant in all tests, including those with multiple comparisons (adjusted $P$ value).

## ACKNOWLEDGMENTS

This study was partially supported by grants PID2021-125801OB-100, PLEC2022-009356, and PDC2022-133577-I00 to E.T., PID2020-112783GB-C21 to A.J., and PID2020-112783GB-C22 to A.R., funded by the Spanish Ministry of Science and Innovation MCIN/AEI/10.13039/501100011033 and "ERDF A way of making Europe," the CERCA programme and AGAUR-Generalitat de Catalunya (2021SGR01545), the European Regional Development Fund (FEDER), and Catalan Cystic Fibrosis association. The project that gave rise to these results received the support of a fellowship from "la Caixa" Foundation (ID 100010434). The fellowship code is "LCF/BQ/DI20/11780040."

The funders had no role in the study design, data collection, and interpretation.

## AUTHOR AFFILIATIONS

[1]Bacterial infections and antimicrobial therapies group, Institute for Bioengineering of Catalonia (IBEC), The Barcelona Institute of Science and Technology, Barcelona, Spain
[2]Microbiology Section, Department of Genetics, Microbiology and Statistics, Faculty of Biology, University of Barcelona, Barcelona, Spain
[3]Molecular Biology Institute of Barcelona (IBMB-CSIC), Barcelona, Spain
[4]Biochemistry and Molecular Biology Department, Universitat Autònoma de Barcelona, Bellaterra, Spain

## AUTHOR ORCIDs

Betsy Verónica Arévalo-Jaimes  http://orcid.org/0000-0002-3363-5403
Inmaculada Ponte  http://orcid.org/0000-0002-9448-6915
Albert Jordan  http://orcid.org/0000-0002-3970-8693
Alicia Roque  http://orcid.org/0000-0002-6206-6481
Eduard Torrents  http://orcid.org/0000-0002-3010-1609

## FUNDING

| Funder | Grant(s) | Author(s) |
| --- | --- | --- |
| Ministerio de Ciencia e Innovación (MCIN) | PID2021-125801OB-100, PLEC2022-009356, and PDC2022-133577-I00 | Eduard Torrents |
| Ministerio de Ciencia e Innovación (MCIN) | PID2020-112783GB-C21 | Albert Jordan |
| Ministerio de Ciencia e Innovación (MCIN) | PID2020-112783GB-C22 | Inmaculada Ponte |
| | | Alicia Roque |
| Generalitat de Catalunya (Government of Catalonia) | 2021SGR01545 | Eduard Torrents |
| 'la Caixa' Foundation ('la Caixa') | LCF/BQ/DI20/11780040 | Betsy Veronica Arévalo-Jaimes |

## AUTHOR CONTRIBUTIONS

Betsy Verónica Arévalo-Jaimes, Conceptualization, Formal analysis, Investigation, Methodology, Software, Writing – original draft, Writing – review and editing | Mónica Salinas-Pena, Investigation, Methodology, Writing – review and editing | Inmaculada Ponte, Conceptualization, Data curation, Funding acquisition, Investigation, Methodology, Resources, Writing – review and editing | Albert Jordan, Conceptualization, Data curation, Funding acquisition, Supervision, Writing – review and editing | Alicia Roque,

Conceptualization, Funding acquisition, Investigation, Methodology, Supervision, Writing – original draft, Writing – review and editing | Eduard Torrents, Conceptualization, Data curation, Formal analysis, Funding acquisition, Investigation, Methodology, Project administration, Resources, Supervision, Validation, Writing – original draft, Writing – review and editing

## ADDITIONAL FILES

The following material is available online.

### Supplemental Material

**Supplemental Material (mSystems00704-24-s0001.docx).** Figure S1.

### Open Peer Review

**PEER REVIEW HISTORY (review-history.pdf).** An accounting of the reviewer comments and feedback.

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
