## [Reviewer comments · mSystems]

Antimicrobial and antibiofilm activity of human recombinant H1 histones against bacterial infections

Besty Verónica Areválo, Mónica Salinas-Pena, Immaculada Ponte, Albert Jordan, Alicia Roque, and Eduard Torrents

Corresponding Author(s): Eduard Torrents, Institut de Bioenginyeria de Catalunya

Review Timeline:

Submission Date:	May 20, 2024
Editorial Decision:	September 4, 2024
Revision Received:	September 17, 2024
Accepted:	September 26, 2024

Editor: Adina Howe

Reviewer(s): Disclosure of reviewer identity is with reference to reviewer comments included in decision letter(s). The following individuals involved in review of your submission have agreed to reveal their identity: Mujeeb Ur Rehman (Reviewer #1)

Transaction Report:

DOI: <https://doi.org/10.1128/msystems.00704-24>

Re: mSystems00704-24 (Antimicrobial and antibiofilm activity of human recombinant H1 histones against bacterial infections)

Dear Dr. Eduard Torrents:

I agree with the reviewers on the impact of this work. I also believe their comments for clarification of minor points in the manuscript as well as making tables and figures will improve the impact of this publication.

Revision Guidelines

Sincerely,
Adina Howe
Editor
mSystems

Reviewer #1 (Comments for the Author):

Overall, this study presents promising results regarding the antimicrobial potential of histones, particularly H1.0 and H1.4, in combating infections caused by *P. aeruginosa*. However, further research is needed to fully understand the mechanisms of action and potential clinical applications of histones as antimicrobial agents.

Reviewer #2 (Comments for the Author):

The manuscript by Arévalo-Jaimes et al. describes the potential antibiofilm activity of histones against *P. aeruginosa*, and their possible mechanism of action. In particular, the authors focused on H1.0, H1-0 C-terminal domain (CTD), H1.2 and H1.4, and their combined activity with ciprofloxacin and gentamicin.

Although the experiments are consistent, all data have been obtained using the common laboratory reference strain PAO1, which is moderately virulent and susceptible to ciprofloxacin, thus, the activity of ciprofloxacin itself might have masked the real antimicrobial effect of histone proteins. Corroborating the results using clinical isolates resistant to Cpx or other *P. aeruginosa* reference strains can strengthen the data on the suggested antimicrobial activity of histones.

Experiments with gentamicin are reported only in Table 2, whereas no further investigations are described, especially against biofilms. Having ciprofloxacin and gentamicin have different mechanisms of action, it would be interesting to duplicate what the authors carried out with Cpx with Gm. Moreover, as the tolerance to antimicrobials often depends on growth conditions, I would suggest considering microaerophilic conditions besides aerobic ones.

As they stand, data on *E. coli* and *S. aureus* are irrelevant, and it is unclear what in vivo data in Table 1 refers to (*P. aeruginosa* infections?). Concerning *E. coli*, MICs are almost doubled compared to *P. aeruginosa*, could the authors comment on that?

Minor comments:

Line 96: I suggest specifying "INVERTEBRATE animal model *Galleria mellonella*".

Line 169: add the p value instead of "statistically significant".

Line 195: how did the author establish Cpx dosage?

Line 344: specify the histone protein concentrations that have been tested

Line 344: TSB medium instead of media

Reviewer #3 (Comments for the Author):

The study presents valuable insights into the antimicrobial and antibiofilm activities of histone H1 subtypes, and the findings are both novel and relevant to the field of antimicrobial research.

In the methods:

- Histone expression and purification process could be described with more precision, including the specific conditions (e.g., cloning strategy) used.
- In Antibiotic-histones synergy testing by checkerboard assay, It would be helpful to specify the range or fold dilutions of the histone and antibiotic concentrations used on each axis of the microtiter plate.
- Clarify the purpose and outcomes of the double injection control group in the results section. It would also be beneficial to discuss how the findings from this control were used to interpret the efficacy and toxicity results.
- Some details of the methodology, such as the specific conditions under which the larvae were maintained (e.g., temperature, humidity) and how the infection was administered (e.g., route of injection, volume), are not mentioned. This information would help others to further replicate your results.
- The choice of *Galleria mellonella* as the in vivo model is mentioned, but the rationale for its selection over more complex mammalian models could be explained
- The methods for statistical analysis are briefly mentioned, but the section could be expanded for robustness. There is no mention of whether the data were tested for normality or homogeneity of variance, which are prerequisites for the valid application of ANOVA. If multiple tests were conducted, it's important to mention whether any corrections for multiple comparisons were applied, such as Bonferroni correction. The section mentions a p-value <0.05 as the threshold for statistical significance, but it does not explain whether this threshold was adjusted for multiple comparisons or whether other statistical metrics, such as confidence intervals, were used. The section does not describe how missing data, outliers, or other potential issues were handled. This is important for understanding the robustness of the statistical analysis.

In the results:

- Line 129, "Noteworthy, upon incubation with *P. aeruginosa* PAO1, H1 histones apparently suffered a partial cleavage..." - clarification of what is apparent or observed would or how it is supported by the data would be helpful
- In figure 2, was the presence of smaller histone forms in the pellet fraction quantified, and could this cleavage be linked to specific bacterial proteases?
- In figure 3, the membrane damage observed in *P. aeruginosa* PAO1 after histone treatment is noticeable, what percentage of cells exhibited membrane damage, and how was this determined?
- Figure 5 - The points of measurements and variation between replicates is not clear here - it is unclear generally if enough statistical power was obtained given the small sample size as shown. Suggest adding error bars.

Discussion

- It would be helpful to have more discussion on the discrepancy between the in vitro synergy observed between H1.0 and ciprofloxacin and the lack of synergy in vivo.
- It would increase impact to address how the findings might translate to clinical settings. Suggestions for how histones could be formulated or delivered in humans or challenges would enhance the practical relevance of the study.

September 17th, 2024

Dear Prof. Adina Howe,

Thank you for the thorough review of our paper and the opportunity to submit a revised version. We much appreciate the reviewer's constructive comments on our manuscript (Manuscript Number: **mSystems00704-24**, *Antimicrobial and antibiofilm activity of human recombinant H1 histones against bacterial infections*, which have been of great help and have greatly improved the manuscript over the previous version.

Our responses to his/her comments are detailed below (in red).

With the manuscript changes detailed below and our answers to the reviewer's comments, we hope you will now find the revised version of our manuscript acceptable for publication in mSystems.

Sincerely,

Dr Eduard Torrents

Response to reviewers

Reviewer #2 (Comments for the Author):

The manuscript by Arévalo-Jaimes et al. describes the potential antibiofilm activity of histones against *P. aeruginosa*, and their possible mechanism of action. In particular, the authors focused on H1.0, H1-0 C-terminal domain (CTD), H1.2 and H1.4, and their combined activity with ciprofloxacin and gentamicin. Although the experiments are consistent, all data have been obtained using the common laboratory reference strain PAO1, which is moderately virulent and susceptible to ciprofloxacin, thus, the activity of ciprofloxacin itself might have masked the real antimicrobial effect of histone proteins. Corroborating the results using clinical isolates resistant to Cpx or other *P. aeruginosa* reference strains can strengthen the data on the suggested antimicrobial activity of histones.

Thank you for your comment. We agree with the reviewer that using other clinical isolates would strengthen our observations. However, our current focus is on measuring the susceptibility of histones in a well-characterized Pseudomonas aeruginosa wild-type strain. This strain provides important insights into bacterial susceptibility to various histones. While we recognize that the use of different clinical isolates is crucial for assessing the potential application of histones as antimicrobial agents in clinical settings, this would be more relevant in future studies involving different animal infection models and bacterial strains with varied antibiotic resistance patterns.

Experiments with gentamicin are reported only in Table 2, whereas no further investigations are described, especially against biofilms. Having ciprofloxacin and gentamicin have different mechanisms of action, it would be interesting to duplicate what the authors carried out with Cpx with Gm.

Thank you for the comment. We agree with the reviewer on the importance of using two antibiotics with different mechanisms of action, which is why we tested the synergy between Cpx and Gm (see Table 2). However, since we did not observe any improvement when combining Gm with histones (See Table 2), we focused our time and efforts on Cpx. In addition to inhibiting protein synthesis, we found that Gm also interacts with LPS (Yokota et al., 2018), likely competing with histones. We agree with the reviewer's observation, and because we consider this an important point for the reader, we have included the corresponding explanation in the discussion section, lines 309-315, along with an additional reference.

Yokota, S. I., Hakamada, H., Yamamoto, S., Sato, T., Shiraishi, T., Shinagawa, M., & Takahashi, S. (2018). Release of large amounts of lipopolysaccharides from Pseudomonas aeruginosa cells reduces their susceptibility to colistin. International journal of antimicrobial agents, 51(6), 888-896.

Moreover, as the tolerance to antimicrobials often depends on growth conditions, I would suggest considering microaerophilic conditions besides aerobic ones.

The reviewer is correct. We did not assess microaerophilic conditions, as biofilms inherently represent a mode of life where oxygen gradients are present. This suggests that microaerophilic conditions could be naturally represented within this form of growth.

As they stand, data on *E. coli* and *S. aureus* are irrelevant, and it is unclear what in vivo data in Table 1 refers to (*P. aeruginosa* infections?).

Thank you for the comment. We agree with the reviewer that initial exploration of histones activity in the other bacterial species shows a broad-spectrum antimicrobial effect of histones that must be highlighted. Thus, we have been added a small sentence in the discussion section lines 219-220. Regarding in vivo data, we decided to remove this information from the table because was causing confusion. As they were referred to toxicity data of histones, we relocated the information to the corresponding results section (line 191).

Concerning *E. coli*, MICs are almost doubled compared to *P. aeruginosa*, could the authors comment on that?

*Thank you for the comment. Our results align with the study by Jodoin & Hincke (2018), where the MIC values of histone H5 from chicken (analogous to human H1) were ~5 µg/mL for *E. coli* strains compared to ~3 µg/mL for *P. aeruginosa*.*

*We speculate that this difference may be associated with the membrane and cell wall characteristics for each species. For example, structural diversity in the LPS, such as covalent modifications in the core oligosaccharide, lipid A or antigen O, can differ among Gram-negative bacteria. As a result, different immunogenic processes have been reported to be elicited by the LPS antigens of *E. coli* and *P. aeruginosa* (Pandur et al., 2021). Thus, histone binding and their antimicrobial effect could also be influenced by species-specific variations.*

Jodoin, J., & Hincke, M. T. (2018). Histone H5 is a potent antimicrobial agent and a template for novel antimicrobial peptides. Scientific reports, 8(1), 2411.

Pandur, E., Tamási, K., Pap, R., Jánosa, G., & Sipos, K. (2021). Distinct effects of Escherichia coli, Pseudomonas aeruginosa and Staphylococcus aureus cell wall component-induced inflammation on the iron metabolism of THP-1 cells. International Journal of Molecular Sciences, 22(3), 1497.

Minor comments:

Line 96: I suggest specifying "INVERTEBRATE animal model Galleria mellonella".

Thank you for the suggestion. We have included the clarification in the correspondent line. See line 97.

Line 169: add the p value instead of "statistically significant".

Thank you for the suggestion. We have made the respective change in line 173.

Line 195: how did the author establish Cpx dosage?

We used a 20 mg/kg dosage of Cpx for Galleria mellonella larvae treatment (positive control), considering that the standard treatment for a human adult consists of 40 mg/kg daily doses, divided into two 20 mg/kg intakes every 12 h. We then tested serial dilutions down to 1.25 mg/kg. This dosage was selected because it provided partial protection to the larvae, allowing us to evaluate potential improvements in Cpx treatment through synergism.

Line 344: specify the histone protein concentrations that have been tested

Thank you for the suggestion. We have included the clarification in the correspondent line (now 384).

Line 344: TSB medium instead of media

Thank you for the correction. We have made the respective change (see line 384).

Reviewer #3 (Comments for the Author):

The study presents valuable insights into the antimicrobial and antibiofilm activities of histone H1 subtypes, and the findings are both novel and relevant to the field of antimicrobial research.

Thank you for your comment.

In the methods:

- Histone expression and purification process could be described with more precision, including the specific conditions (e.g., cloning strategy) used.

Thank you for the comment. We have included the required information in lines 333-341 and 343-352.

- In Antibiotic-histones synergy testing by checkerboard assay, It would be helpful to specify the range or fold dilutions of the histone and antibiotic concentrations used on each axis of the microtiter plate.

Thank you for the comment. We have included the required information in lines 410-411.

- Clarify the purpose and outcomes of the double injection control group in the results section. It would also be beneficial to discuss how the findings from this control were used to interpret the efficacy and toxicity results.

Thank you for this observation. Doble injection groups are controls to discard mortality associated with needle injuries during both the infection and treatment step. We included a PBS – Histone (20 mg/kg) group in the toxicity assessment to eliminate any adverse events in larvae previously subjected to stress from the injection procedure. The lack of morbidity/mortality in this group indicated it was safe to use histones as treatment in a double injection procedure.

Additionally, the lack of mortality in the PBS – PBS group during the antimicrobial efficacy experiment confirmed that observed mortality in other groups was related to bacterial infection and not to larvae injuries from the injection process.

Accordingly, we have modified the lines 477-480 and 487-489 of the methods section and lines 192-196 and 202-204 of the results section to include these clarifications.

- Some details of the methodology, such as the specific conditions under which the larvae were maintained (e.g., temperature, humidity) and how the infection was administered (e.g., route of injection, volume), are not mentioned. This information would help others to further replicate your results.

Thank you for the comment. We have included the required information in lines 469-472, 473-474 and, 479-480.

- The choice of *Galleria mellonella* as the in vivo model is mentioned, but the rationale for its selection over more complex mammalian models could be explained

Thank you for the suggestion. We have included the rationale behind using this animal model of infection. See new information in lines 292-294 of the discussion section.

- The methods for statistical analysis are briefly mentioned, but the section could be expanded for robustness. There is no mention of whether the data were tested for normality or homogeneity of variance, which are prerequisites for the valid application of ANOVA. If multiple tests were conducted, it's important to mention whether any corrections for multiple comparisons were applied, such as Bonferroni correction. The section mentions a p-value <0.05 as the threshold for statistical significance, but it does not explain whether this threshold was adjusted for multiple comparisons or whether other statistical metrics, such as confidence intervals, were used. The section does not describe how missing data, outliers, or other potential issues were handled. This is important for understanding the robustness of the statistical analysis.

Thank you for the comment. We have expanded the "statistical analysis" section including the required information. We hope the robustness of the data analysis is now clearer. See lines 493-500.

In the results:

- Line 129, "Noteworthy, upon incubation with *P. aeruginosa* PAO1, H1 histones apparently suffered a partial cleavage..." - clarification of what is apparent or observed would or how it is supported by the data would be helpful

Thank you for the comment. We have modified lines 129-133 to include the required information suggested. We hope the meaning is now clearer.

- In figure 2, was the presence of smaller histone forms in the pellet fraction quantified, and could this cleavage be linked to specific bacterial proteases?

Unfortunately, bands were not quantified in the experiments in Figure 2. However, considering the volumes, we estimate that the cleaved form represents approximately 10-40% of the total H1, depending on the H1 variant. We believe the reviewer's reasoning is correct and that bacterial proteases are responsible for the histone's cleavage. We have included a small sentence in line 133-134.

- In figure 3, the membrane damage observed in *P. aeruginosa* PAO1 after histone treatment is noticeable, what percentage of cells exhibited membrane damage, and how was this determined?

Thank you for the comment. The percentage of bacteria cells with membrane damage is around 5 - 8.5%, depending on the histone. We determined these values by manually counting damaged cells and dividing by the total number of cells (automated particle counting) from three different replicates using 63x magnification field and ImageJ software. The corresponding information was added to the manuscript in lines 140 and 443-445.

- Figure 5 - The points of measurements and variation between replicates is not clear here - it is unclear generally if enough statistical power was obtained given the small sample size as shown. Suggest adding error bars.

Thank you for the comment. The actual graph is representative of three independent experiments (8 larvae each one). It corresponds to a survival curve, which is the type of plot for this kind of analysis. Unfortunately, the software used for this specific plot (GraphPad Prism v11) does not allow to insert error bars. However, the statistical analysis of this type of survival curves shown in Fig. 5 (log-rank test) are shown in the plot and clearly support our results. If error bars are required, we can provide a manual representation using an xy graphic and put the statistical significance derived from the original graph. The result is the following:

In our opinion, the original figure without error bars is easier to interpret, as is the typical representation of survival curves. Therefore, we would prefer to keep it in this format.

Discussion

- It would be helpful to have more discussion on the discrepancy between the in vitro synergy observed between H1.0 and ciprofloxacin and the lack of synergy in vivo.

Thank you for the comment. We have included additional information in lines 316-323 of the discussion section and hope now clarifies the discrepancy.

- It would increase impact to address how the findings might translate to clinical settings. Suggestions for how histones could be formulated or delivered in humans or challenges would enhance the practical relevance of the study.

Thank you for the suggestion. We agree with the reviewer, and we have expanded the discussion regarding this topic in lines 299-308.

Re: mSystems00704-24R1 (Antimicrobial and antibiofilm activity of human recombinant H1 histones against bacterial infections)

Dear Dr. Eduard Torrents:

Your manuscript has been accepted, and I am forwarding it to the ASM production staff for publication. Your paper will first be checked to make sure all elements meet the technical requirements. ASM staff will contact you if anything needs to be revised before copyediting and production can begin. Otherwise, you will be notified when your proofs are ready to be viewed.

Sincerely,
Adina Howe
Editor
mSystems

Reviewer #2 (Comments for the Author):

The authors answered all the raised issues and clarified the experimental design, methods, and discussion section.